# Cyclic Metronomic Chemotherapy for Pediatric Tumors: Six Case Reports and a Review of the Literature

**DOI:** 10.3390/jcm11102849

**Published:** 2022-05-18

**Authors:** Benjamin Carcamo, Giulio Francia

**Affiliations:** 1Department of Pediatric Hematology Oncology, El Paso Children’s Hospital, El Paso, TX 79905, USA; 2Department of Pediatrics, Texas Tech University Health Science Center, El Paso, TX 79430, USA; 3Border Biomedical Research Center, University of Texas at El Paso (UTEP), El Paso, TX 79968, USA

**Keywords:** metronomic chemotherapy, pediatric tumors, cyclophosphamide, etoposide, valproic acid

## Abstract

We report a retrospective case series of six Hispanic children with tumors treated with metronomic chemotherapy. The six cases comprised one rhabdoid tumor of the kidney, one ependymoma, two medulloblastomas, one neuroblastoma, and a type II neurocytoma of the spine. Treatment included oral cyclophosphamide daily for 21 days alternating with oral etoposide daily for 21 days in a backbone of daily valproic acid and celecoxib. In one case, celecoxib was substituted with sulindac. Of the six patients, three showed complete responses, and all patients showed some response to metronomic therapy with only minor hematologic toxicity. One patient had hemorrhagic gastritis likely associated with NSAIDs while off prophylactic antacids. These data add to a growing body of evidence suggesting that continuous doses of valproic acid and celecoxib coupled with alternating metronomic chemotherapy of agents such as etoposide and cyclophosphamide can produce responses in pediatric tumors relapsing to conventional dose chemotherapy.

## 1. Introduction

Metronomic chemotherapy involves the continuous low-dose administration of chemotherapeutic drugs, and it is currently undergoing clinical evaluation for the treatment of various pediatric and adult malignancies [1,2,3,4,5,6]. For example, for adult tumors, a phase III clinical trial of metastatic colorectal cancer with maintenance metronomic capecitabine plus bevacizumab was reported, as were results of a phase III trial in early-stage breast cancer treated with bevacizumab plus metronomic cyclophosphamide and capecitabine [7,8]. For pediatric tumors, a number of phase II clinical trials have reported promising antitumor activity of metronomic chemotherapy, and these studies include those of Kieran and colleagues [9,10], of Andre and colleagues [11,12,13,14,15,16,17], and of the Baruchel group [18]. For both adult patients and pediatric tumors, the low toxicity that is frequently observed with this treatment strategy coupled with its (often) relatively low costs [4] make it an attractive treatment option. Preclinical studies [2,19,20] have shown that metronomic chemotherapy acts via inhibition of tumor angiogenesis, although additional mechanisms such as activation of the immune system have been proposed for regimens such as low-dose cyclophosphamide [21].

Metronomic chemotherapy may involve monotherapies, e.g., daily oral cyclophosphamide (CTX), but it can also be used to describe metronomic drug cocktails [22,23], including protocols that administer up to five different drugs [10] or more [24,25]. There are also evaluations involving the cyclical use of different drugs, an example of which is the work by Kieran and colleagues [9,10] on pediatric tumors. Here, we describe six children with pediatric cancer treated with cyclic alternating metronomic chemotherapy between 2002 and 2017.

These children had failed standard first line and salvage therapy and had refused or had no access to phase I or II studies. They were the only patients treated with this regimen during this period of time. The intention of the treatment was to provide palliation to children with refractory cancer that had exhausted all the therapeutic options available at the time and when parents requested additional treatment to prolong their children’s life. It was not done to prove the validity of a metronomic regimen. The parents where consented to use a palliative regimen without curative intent so that they would understand the expected side effects of the individual drugs. They understood that it was not known if this treatment would prolong their life or not, and that the palliation was not done in the context of a study.

Metronomic chemotherapy included alternating 21-day cycles of etoposide (VP16) and CTX, along with the continuous administration of celecoxib and valproic acid. Sometimes due to individual reasons only celecoxib or valproic acid was used and sometimes celecoxib was substituted with sulindac. Oral continuous CTX at 50 mg/m^2^/day had previously been used in metronomic regimens with minimal toxicity [26]. Oral VP16 at 50 mg/m^2^/day for 21 days was chosen on the basis of evidence that continuous low-dose etoposide could enhance tumor cytotoxicity while lowering the risk of acute myeloid leukemia [27], and that 21 days of this dose with a 1 week rest had achieved partial response with moderate toxicity in recurrent disseminated medulloblastoma [28]. Alternating cycles of these drugs were chosen to decrease hematologic toxicity from etoposide and the emergence of drug resistance to either drug alone. Dose reduction was allowed for hematologic toxicity. Celecoxib is a selective COX2 inhibitor that has been well tolerated without cardiac toxicity at a dose of 2–16 mg/kg/day for prevention of adenomatous polyps in children [29]. The COX2 pathway is known to be involved in angiogenesis [30], and inhibition of angiogenesis has been reported to produce enhance the antitumor effects of metronomic chemotherapy [31], for which celecoxib was chosen at a dose of 250 mg/m^2^/dose twice a day (equivalent to 8 mg/kg/day). Valproic acid (VA) is a histone deacetylase (HDAC) inhibitor with a long safety record in children with seizures that has shown antitumor effects in preclinical models of medulloblastoma, for which it was chosen at the standard pediatric dose of 15 mg/kg/day in two divided doses [32,33]. Furthermore, a major reason for the choice of these drugs was also their low toxicity profile, low cost, and availability. Medications were administered as capsules (the exception was for patient 3, for whom the liquid preparation of etoposide was mixed with cranberry juice). Our results suggest that this approach showed promising antitumor activity in six pediatric cases, and we propose that a cyclical regimen of 21 day VP16 followed by 21 day oral CTX on a backbone of continuous celecoxib and VA be further evaluated in these settings.

### 1.1. Patient 1

Patient 1 was a Hispanic boy who presented with metastatic rhabdoid tumor of the kidney at 34 months of life (Figure 1). He had a 7 cm × 11 cm mass in the left kidney, retroperitoneum invasion, and extension into the renal vein and inferior vena cava. He had metastasis to retroperitoneal lymph nodes and lungs, as well as rare micro-aggregates of cohesive abnormal large mononuclear cells suggestive of metastatic malignant tumor in the bone marrow by morphology alone (INH1 was not done). Bone marrow karyotype was 46,XY with no abnormal clones. He received vincristine–irinotecan, followed by cyclophosphamide–doxorubicin, followed by high-dose methotrexate as per regimen UH1 of AREN0321. This was followed by radical left nephrectomy at week 15. Subsequent evaluation revealed a residual 2 mm lung nodule but otherwise complete response. He subsequently received radiation to lungs and abdomen, followed by cycles of cyclophosphamide–carboplatin–etoposide and vincristine–doxorubicin–cyclophosphamide. Evaluation at the end of this therapy revealed recurrent disease to the lungs (two nodules measuring 3 mm and 1.5 mm in diameter) and an abnormal bone marrow clone, t(5;17)(q13;p13) in 15% of the cells. This cytogenetic abnormality has been previously reported in treatment-related myeloid neoplasia [34], and its presence was worrisome for early development of this condition from conventional chemotherapy and radiation, but there was no morphologic evidence of this entity at that time. No additional analysis was carried out, such as evaluation of INI1 loss. He was subsequently started on a metronomic regimen of VP16–CTX–celecoxib, and this maintenance regimen was effective at controlling the disease with no recurrent rhabdoid tumor or changes in the pulmonary nodules. Metronomic doses were as follows: VP16 37.5 mg/m^2^ PO daily days 1–21, followed CTX 50 mg/m^2^ PO daily days 22–42. Celecoxib was administered at 250 mg/m^2^ PO twice a day and VA 15 mg/kg/day divided into two doses (dose adjusted to achieve levels of 100–150 ug/mL). He subsequently developed expansion of the original bone marrow clone (5;17)(q11.2p11.2) to 80% of metaphase cells, for which metronomic therapy was changed to bevacizumab (5 mg/kg every 3 weeks), temozolomide (60 mg/m^2^/day on days 1–21), CTX (25 mg/m^2^/day on days 22–42), VA (15 mg/kg/day divided into two doses) and celecoxib (250 mg/m^2^ PO twice a day) after 17 months of treatment. He subsequently developed myelodysplastic changes consistent with treatment-related myeloid neoplasia, for which temozolomide and CTX were stopped 12 months later. These bone marrow myelodysplastic changes eventually progressed, in the absence of evidence of rhabdoid tumor reappearance, for which celecoxib–VA–bevacizumab were stopped, and he had a related donor allogenic bone marrow transplant 4 months later. He subsequently died of transplant-related infectious complications. He did not have evidence of recurrent rhabdoid tumor according to imaging studies at the time of his death. Autopsy was offered and declined. The rhabdoid tumor of this patient effectively responded to the metronomic chemotherapy for a period of 3 years.

### 1.2. Patient 2

Patient 2 was diagnosed with anaplastic ependymoma when he presented with right hemianopsia, severe vision loss, and a 12 cm × 8 cm × 8 cm left temporo-occipital tumor at 6 years of life. He had two consecutive surgeries to remove the tumor, but a 1 cm unresectable lesion was left behind. He then received 5940 cGy field radiation to the tumor bed. Three months after radiation, he developed seizures, transient right arm paralysis, and recurrent disease to the left cerebral hemisphere, for which he received vincristine, carboplatin, and cyclophosphamide following the guidelines of ACNS0121 [35]. Three months later, he developed progressive disease with a large infiltrative enhancing mass (~5 cm) in the left temporal, occipital, and parietal lobes with involvement of the left thalamus, internal capsule, left external capsule, and insula, as well as leptomeningeal metastasis to the thoracic spine. He had no access to Phase I or II studies and was unable to travel for lack of insurance and migratory status, for which treatment was changed to palliation chemotherapy with oral VP16 (50 mg/m^2^/day for 21 days) alternating with oral CTX (50 mg/m^2^/day for 21 days) and continuous celecoxib (120 mg/m^2^ PO bid) and VA. Celecoxib was subsequently changed to sulindac (8 mg/kg/day) due to drug access issues. He had an excellent response to palliation with resolution of all enhancing lesions and no evidence of residual disease after 2 years of therapy. (Figure 2). About this time VA was added to his seizure disorder management and continued until his last visit. He received 18 additional months of metronomic chemotherapy after complete response. He had no evidence of recurrent disease when he was lost to follow up 8 years from initial diagnosis and 4 years off metronomic chemotherapy. This patient showed complete response to the metronomic regimen.

### 1.3. Patient 3

Patient 3 was a Hispanic girl diagnosed with medulloblastoma, when she presented with ataxia, dysmetria, increased head circumference and hydrocephalus at 10 months of age. She had gross total resection of a 5 × 5 × 4.5 cm posterior fossa tumor. Postoperative MRI revealed gross total resection and partial decompression of the fourth, third, and lateral ventricles. Following surgery, imaging studies showed no evidence residual disease or leptomeningeal involvement. However, ventricular fluid had evidence of microscopic disease with a five-cell cluster of malignant cells. Cerebrospinal fluid was negative. She received induction chemotherapy with cisplatin, cyclophosphamide, vincristine, and oral etoposide, followed 4 months later by radiotherapy to the posterior fossa, followed by another 8 months of chemotherapy as per POG 9631 [36]. Six months after therapy was completed, microscopic relapse was documented in cerebrospinal fluid with three clusters of malignant cells. Imaging studies failed to document gross leptomeningeal involvement. She started metronomic chemotherapy for relapsed medulloblastoma with 21 days of oral VP16 (35 mg/m^2^/day) every 28 days. After 7 months of oral VP16, persistent microscopic disease was still present in the cerebrospinal fluid, and VA was added to the treatment. She achieved remission 5 months later, and serial cerebrospinal fluids remained consistently negative since then. She received a total of 39 months of metronomic chemotherapy including 21 months of etoposide and 31 months of VA. She was free of disease 5 years after metronomic chemotherapy was stopped.

### 1.4. Patient 4

Patient 4 was a Hispanic girl diagnosed with localized medulloblastoma without desmoplastic or anaplastic features when she presented with headaches, emesis, and blurred vision and a 4.9 × 4 × 4.5 cm heterogeneously enhancing tumor filling the fourth ventricle with associated moderate obstructive hydrocephalus at 5 years of age. Following gross total resection of the tumor, residual linear enhancement in the tumor bed was documented. She subsequently received 2340 cGy radiation to the craniospinal axis with a 3060 cGy boost to the posterior fossa (5400 cGy total) with concomitant vincristine after surgery. This was followed by 8 months of maintenance with vincristine, cisplatin, lomustine alternating with vincristine and cyclophosphamide as per ACNS0331 [37].

Three months off therapy, she developed a first relapse (Figure 3A) with recurrent tumors in the anterior horn of the left lateral ventricle (two), posterior horn of the left lateral ventricle (one), and anterior horn of the right lateral ventricle (one). CSF showed one abnormal cell. She was treated with 7 weeks of temozolomide followed by 1400 cGy of radiation to the ventricles and Gamma Knife surgery to a residual nodule, followed by autologous peripheral blood stem-cell transplantation (PBSCT), followed by 6 months of isotretinoin.

Three months later, she experienced a second relapse (Figure 3B) with a 1.7 × 1.4 × 0.5 cm enhancing tumor in the fourth ventricle, for which she was started on metronomic chemotherapy with alternating temozolomide (60 mg/m^2^/day; days 1–21) and CTX (30.9 mg/m^2^/day; days 22–42), daily celecoxib (250 mg/m^2^) and VA (15 mg/kg), and bevacizumab (5 mg/kg) every 3 weeks. After 3 months on metronomic chemotherapy (Figure 3C), a partial response was documented with improved enhancing lesions in the dorsal brainstem and inferior vermis. At 4 months (Figure 3D), she developed multiple cranial nerve neuropathies and heterogeneous enhancement with mild increased volume at the left midbrain at the level of the left cerebellar peduncle, for which temozolomide was changed to VP16 (24.6 mg/m^2^). This was complicated by hematologic toxicity requiring several treatment interruptions and VP16 dose reductions. After 8 months on treatment, MRI showed resolution of the enhancing tumor, an interval decrease in T2 changes, and diffusion restriction in the left cerebellar peduncle (Figure 3E). After 10 months on treatment (Figure 3F), she developed an extensive infiltrative tumor involving the dorsal mesencephalon, pons, and lower brainstem, and she died of her disease 2 months later. This heavily treated patient refractory to aggressive standard salvage therapy experienced 8 months of response and survived 12 months on metronomic chemotherapy.

### 1.5. Patient 5

This patient was a Hispanic boy diagnosed with metastatic neuroblastoma when he presented with an 8 × 12 cm right suprarenal mass, extensive retroperitoneal and mesenteric lymphadenopathy, spinal cord compression, and metastatic bone disease to the right scapula and body of L1 at 9 years of age. He was treated with high-dose cisplatin and etoposide, cyclophosphamide, Adriamycin and vincristine, ifosfamide and etoposide, and carboplatin and etoposide, followed by subtotal resection of the retroperitoneal tumor and right kidney as per ANBL00P1 [38]. PBSCT was not available due to a lack of funds, for which he received two cycles of topotecan–cyclophosphamide, followed by 2160 cGy radiation to the tumor bed and spine, followed by 6 months of isotretinoin. He had no evidence of residual disease at the end of treatment. Seven months later, he developed recurrent neuroblastoma involving the right external iliac lymph nodes, for which he started palliative metronomic chemotherapy with oral VP16 (50 mg/m^2^/day; days 1–21) alternating with oral CTX (75 mg/m^2^/day; days 22–41) and continuous sulindac (4 mg/kg/day). He had hematologic toxicity requiring interruptions and dose reductions of both VP16 and CTX. Evaluation showed decreased tumor size at 5 and 8 months of therapy. He subsequently developed progressive disease after 11 months on therapy, for which sulindac was changed to celecoxib (500 mg/m^2^/day). Two months later while off prophylactic antacids, the patient developed hemorrhagic gastritis, hematemesis, aspiration pneumonia, acute renal failure, and seizures, and died 13 months after starting metronomic chemotherapy.

### 1.6. Patient 6

This patient is a Hispanic girl found to have a large cervicomedullary tumor with a cervico-thoracic central syrinx with tumor seeding when she presented with right shoulder drop and right upper-extremity weakness at 3 years of age (Figure 4A). She had resection of the cervicomedullary portion of the tumor. Evaluation revealed a type 2 neurocytoma of the spine.

She subsequently received intensity-modulated radiotherapy to the tumor bed and residual cervico-thoracic tumor. Following radiation, a residual 11 mm expansile cervicomedullary nodule, centered at the C3–C4 level, was documented. Five months later, she developed progressive disease with increased size of the cervicomedullary nodule to 15 mm and development of a new second focus of enhancement on the ventral aspect of the spinal cord at C2. She was treated with topotecan, ifosfamide and carboplatin (TIP). She had significant hematologic toxicity and infectious complications during this treatment and no change in tumor size after nine cycles (Figure 4B), for which she was changed to metronomic chemotherapy with 21 days of temozolomide (60 mg/m^2^ day) aleternating with 21 days of CTX (50 mg/m^2^), VA (15 mg/kg), celecoxib (250 mg/m^2^/day) and bevacizumab (5 mg/kg IV every 3 weeks). Tumor response was noted 3 months after treatment with decreased tumor size to 9–10 mm (Figure 4C). Tumor size remained stable on subsequent studies, and temozolomide and CTX were stopped at 18 months, celecoxib and VA were stopped at 36 months, and bevacizumab was stopped at 48 months (Figure 4D). Metronomic chemotherapy was restarted after tumor progression was documented 10 months later (Table 1). She developed stable disease and treatment was stopped 15 months later. She remains with stable disease 4 years off therapy.

## 2. Discussion

Here, we report six cases of pediatric malignancies treated with metronomic chemotherapy that included alternating 21 days cycles of oral etoposide and cyclophosphamide in a backbone of celecoxib and/or valproic acid (Table 1). Because metronomic chemotherapy was used for palliation and not in the context of a study, some variation occurred due to medical and social issues such as individual choice, lack of insurance, and migration status. As noted, in some cases, bevacizumab and/or temozolomide were also administered. This treatment strategy produced significant tumor responses, including three complete responses. Three patients had hematologic toxicity with metronomic chemotherapy requiring interruptions and dose reductions. Two of these patients eventually progressed and died of their disease, and it is unclear whether the interruptions interfered with the metronomic effect. Patient 5 had hemorrhagic gastritis presumably from NSAIDs, for which this regimen should be used with prophylactic antacids as per current COG guidelines.

Three patients with brain tumors were treated. Unfortunately, they preceded easy access to molecular studies and, thus, are not reported according to the current WHO classification. The first patient (case 2) that used this regimen was treated in 2002. He had an anaplastic ependymoma that had failed surgery, radiation, and chemotherapy and had progressed with a large hemispheric tumor and spinal cord metastasis. He started treatment with alternating etoposide–cyclophosphamide on a backbone of celecoxib. Valproic acid was later added to the treatment for seizure control. He achieved a complete response at 2 years, stopped therapy 18 months later, and was still in remission 4 years off therapy when lost to follow-up 8 years later. Two patients with medulloblastoma were treated. The first patient relapsed with clusters of malignant cells in CSF with normal imaging studies 6 months off therapy. The malignant cells persisted after 7 months of metronomic etoposide but resolved 5 months after adding valproic acid to the regimen. She remains free of disease 15 years off therapy. Since continuous celecoxib–valproic acid in the context of cyclic chemotherapy was well tolerated and the second patient may have responded to valproic acid, this medication was added to celecoxib in subsequent patients. The second medulloblastoma relapsed with brainstem involvement shortly after completing first- and second-line therapy. She was initially treated with alternating temozolomide–etoposide, bevacizumab, and continuous celecoxib–valproic acid but developed progressive disease 4 months later. She subsequently changed temozolomide to etoposide, achieved partial response 6 months later, and survived 12 months on metronomic chemotherapy. These three cases highlight the potential role of metronomic chemotherapy in refractory brain tumors. Recently, 29 children with refractory medulloblastoma were treated with a metronomic regimen that included alternating etoposide–cyclophosphamide, continuous thalidomide–celecoxib–fenofibrate, bevacizumab, and intraventricular liposomal cytarabine–etoposide (MEMMAT) [39]. EFS was 33% and 28% at 5 and 10 years, while OS was 44% and 39% at 5 and 10 years, respectively [39]. These results make MEMMAT the treatment of choice in refractory medulloblastoma. This regimen has some elements included in our regimen. For example, it contains alternating etoposide–cyclophosphamide and continuous celecoxib but lacks valproic acid. Since it is difficult to tell what elements in a complex metronomic regimen add to the activity, it is reasonable to explore our less complex regimen in patients that refuse invasive interventions or have poor access to care.

One patient with refractory metastatic neuroblastoma was treated after failing first-line therapy. He had no funds and was not able to travel to pursue second-line treatments including stem-cell transplant. He was treated with alternating etoposide–cyclophosphamide and sulindac. He had a partial response at 5 months but progressed at 11 months. Since then a regimen including cycles of 4 days of rapamycin–dasatinib followed by 5 days of irinotecan–temozolomide (RIST) showed impressive responses in refractory neuroblastoma with 90% response in 21 children including 12 CR and three PR, with 43% OS at 143 weeks [40]. While these results are impressive, rapamycin and dasatinib may not be readily available to patients with poor access to care. Most recently, 167 patients with high-risk stage 4 neuroblastoma without access to autologous stem-cell transplantation or anti-GD2 antibody therapy were treated with or without a metronomic regimen including continuous etoposide, cyclophosphamide, vinorelbine, topotecan, and celecoxib, including 106 patients in the metronomic arm and 61 in the non-metronomic arm. The 3 years of event-free survival (EFS) was 42.5% versus 29.4%, and overall survival (OS) was 71.1% versus 59.4%, respectively [41]. While the Corbacioglu et al. [40] report strongly supports RIST for refractory neuroblastoma, the report from Sun et al. [41] showing a benefit of oral metronomic chemotherapy comparable to autologous stem-cell transplant and anti-GD2 in high-risk Neuroblastoma suggests a possible role for metronomic chemotherapy in refractory neuroblastoma. Metronomic chemotherapy either with continuous drug delivery as reported by Sun et al. [41] or with alternating drugs as in this report needs to be further evaluated in these patients.

This report and previous reports by Kieran et al. [9,10], Andre et al. [11,12,13], Sharp et al. [18], Slavc et al. [39] and Sun [41], highlight possible metronomic regimen strategies for the treatment of refractory pediatric cancer. These metronomic strategies are summarized in Table 2.

We have previously used molecular genetics to identify a driving metabolic pathway within refractory Inflammatory Myofibroblastic Tumor that was specifically targeted with the TKI Lorlatinib with good response [42]. We also used proteomics in a refractory childhood embryonal tumor with multilayered rosettes of the brain to identify a driving metabolic pathway that was targeted with the TKI desatinib that and resulted in gross total resection of the tumor and prolong survival [43]. Unfortunately we have also found that these heavily pretreated tumors frequently become resistant to TKI after a few months of response [42]. One way to overcome this difficulty could be to add a backbone of metronomic chemotherapy that would target the micro environment, malignant angiogenesis and drug resistance, while the TKI would target aberrant metabolic pathways in the tumor, increase chemotherapy susceptibility and induce apoptosis.

While it has been noted that a VEGFR targeting TKI may blunt the ability of metronomic chemotherapy [44] to activate the immune system (suggesting antagonistic activity), we have previously shown that a metronomic chemotherapy plus a neutralizing anti-VEGFR antibody preclinically delays the growth of B16 melanoma lung metastases [45]. In addition, effective combination of the TKI sorafenib with metronomic chemotherapy has also been reported preclinically [46].

Collectively, these observations have led us to propose a study that incorporates tyrosine kinase inhibitors (TKI) to metronomic chemotherapy. Our backbone of choice to compare with TKI is celecoxib-valproic acid based on our experience and the growing interest in anti-histone drugs that have undergone trials with responses in difficult pediatric tumors [47]. All patients will receive alternating cyclophosphamide and etoposide as described in this report. This study depicted in Figure 5 will involve screening refractory patient’s tumors for metabolic targets, and treating the patients with alternating etoposide-cyclophosphamide and either continuous celecoxib-valproic acid or continuous TKI if a targetable aberrant metabolic pathway is identified.

The mechanisms via which metronomic chemotherapy produces antitumor effects are numerous [1] and are yet to be fully elucidated. Thus, in 2003, in collaboration with Bocci and colleagues [19], we reported that continuous low-dose CTX (as well as other drugs such as paclitaxel) can induce endothelial cells to overexpress the angiogenesis inhibitor protein thrombospondin-1. These data provide further evidence [48] for the antiangiogenic activity of metronomic chemotherapy. Other mechanisms include an alteration of intratumoral blood flow as observed following the administration of metronomic oral gemcitabine (LY2334737) as we previously observed in a preclinical model [20]. They also include the suppression of the homing of bone marrow-derived endothelial progenitor cells to the periphery of a tumor mass, as shown by Shaked and colleagues [49]. Additional mechanisms of action of metronomic chemotherapy, at least for drugs such as CTX, include immune activation, which likely occurs via suppression of regulatory T cells [21]. Furthermore, in collaboration with Emmenegger and colleagues [50], we reported that preclinical selection of tumors resistant to high-dose chemotherapy does not impair their response to subsequent metronomic regimens of the same drug [50]. These data suggest that mechanisms of tumor resistance to metronomic chemotherapy show little or no overlap with those for the same chemotherapeutics agent (e.g., CTX) given at pulsatile maximum tolerated doses. These results also suggest that a tumor relapsing to pulsatile high dose of a drug such as CTX may still show response to the same drug given on a metronomic schedule.

Metronomic chemotherapy has undergone a number of refinements since it was coined in 2000, to describe preclinical studies with drugs used in a continuous low-dose (as an antiangiogenic strategy or as a maintenance strategy), namely, CTX or vinblastine [48,51]. Since then, the concept of metronomic chemotherapy has evolved for the treatment of adult tumors, as well as in its application in pediatric oncology. Thus, for adult tumors, recent phase III clinical data in metastatic colorectal cancer showed improved progression-free survival in patients receiving maintenance low-dose capecitabine plus bevacizumab [7]. In addition, safety of administration was reported from a phase III clinical trial evaluating bevacizumab plus metronomic cyclophosphamide and capecitabine as first-line therapy in patients with HER2-negative advanced-stage breast cancer [8,52]. On the other hand, for pediatric tumors, reported protocols (as evident from a Pubmed search with the terms “metronomic” and “pediatric”) include the long-term oral administration of daily low-dose mercaptopurine and weekly low-dose methotrexate for children with ALL [11]. They also include a metronomic vincristine/CTX/methotrexate/VA regimen given to children with refractory cancer of various tumor types [53]. Recently, there have been additional reports of metronomic chemotherapy improving survival in high-risk neuroblastoma patients [41], as well as other pediatric malignancies including Ewing sarcoma, osteosarcoma, and rhabdomyosarcoma [15], and studies have confirmed the low toxicity associated with this approach [54]. Alternating VP16 and cyclophosphamide metronomic regimens, with daily celecoxib, were combined with sirolimus in a phase I study of 18 pediatric relapsed or refractory solid and brain tumors, and the combination was found to be well tolerated [55]. It should be noted that among the first studies published on using chemotherapy metronomically for pediatric tumors were those of Kieran and colleagues [10] and of Andre and colleagues [13]. Thus, Kieran’s group originally proposed continuous oral thalidomide and celecoxib with alternating oral etoposide and cyclophosphamide every 21 days. Recently, the same group reported a more complex strategy involving five drugs including metronomic CTX and etoposide, as shown in Table 2. Andre et al. proposed 2 weeks of metronomic etoposide followed by 2 weeks of metronomic CTX, plus celecoxib [13]. Because of the number of drugs involved in some regimens such as the one proposed by Kieran and colleagues (see Table 2), additional studies may be necessary to compare different permutations of the use of such drugs and the different orders in which such drugs are administered metronomically. Such studies may uncover whether specific combinations are important (e.g., CTX plus VA) or if the exact order of administration is critical (e.g., etoposide before CTX, or vice versa). In that respect, the six cases we report add data to a regimen of etoposide, CTX, and VA–celecoxib. Interestingly, with regard to VA–celecoxib as a backbone, we previously reported [20] that metronomic oral gemcitabine (LY2334737) has antitumor effects in preclinical mouse xenograft models. The LY2334737 oral gemcitabine prodrug is, once injected, metabolized into gemcitabine plus VA, suggesting that a metronomic oral gemcitabine schedule may unintentionally result in continuous administration of two drugs (gemcitabine and VA), although this possibility remains to be tested. Overall, these clinical and preclinical observations suggest that metronomic alternating regimens combined with continuous administration of celecoxib and VA have promising activity in tumors relapsing to pulsatile high-dose chemotherapy.

## Figures and Tables

**Figure 1 jcm-11-02849-f001:**
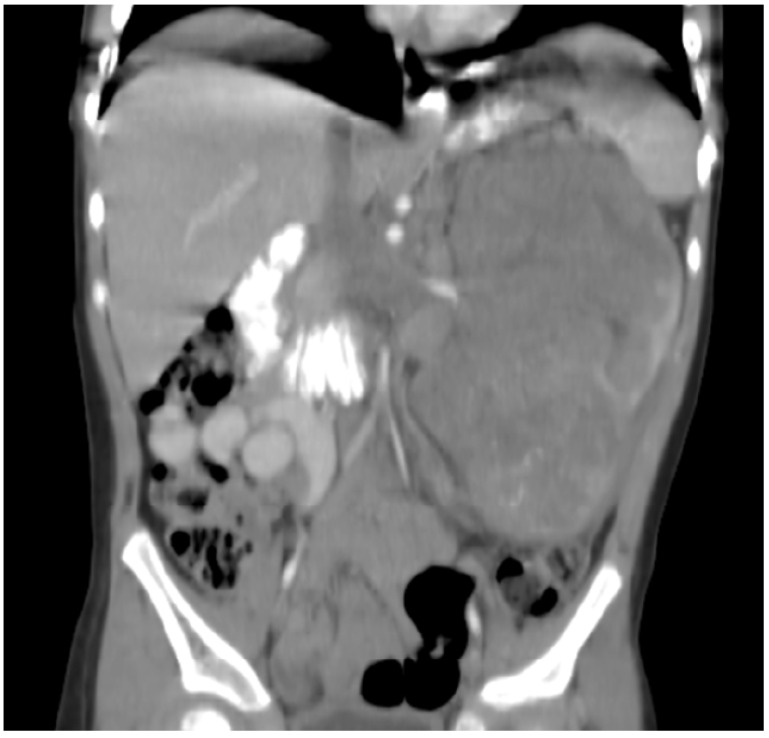
Patient 1: CT with contrast shows a large 11 × 7 cm left renal tumor with retroperitoneal infiltration, regional metastatic retroperitoneal adenopathy, and extension to renal vein and inferior vena cava. The patient also had innumerable solid circumscribed masses throughout the lung parenchyma bilaterally (not shown).

**Figure 2 jcm-11-02849-f002:**
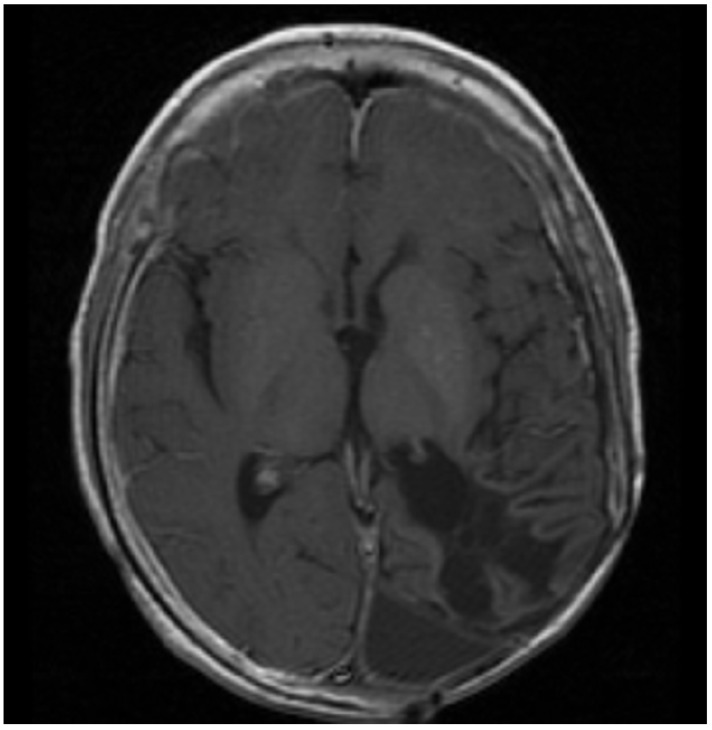
Patient 2: MRI shows confluent encephalomalacia gliosis in the left temporal and occipital lobes. There was no evidence of mass or pathologic enhancement 8 years from diagnosis and 4 years off therapy.

**Figure 3 jcm-11-02849-f003:**
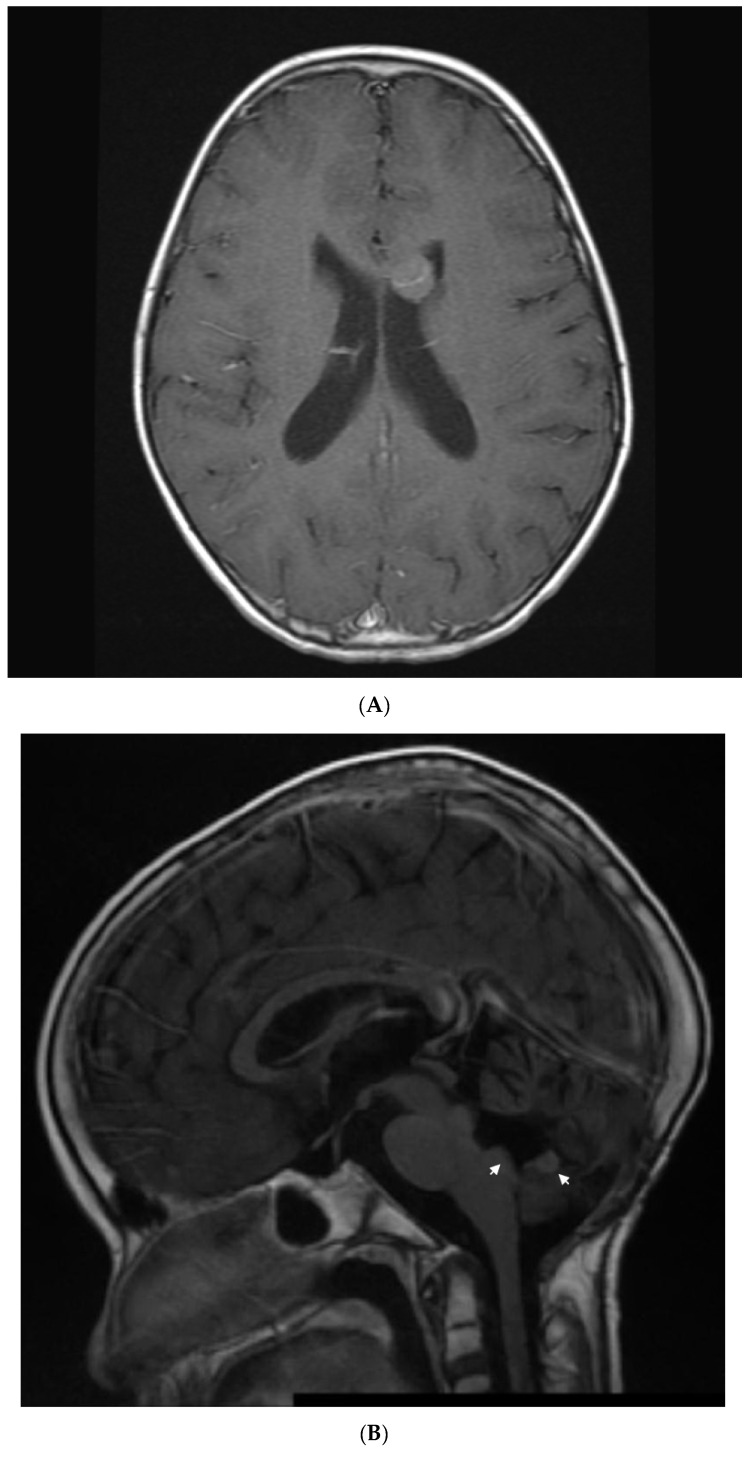
Patient 4: (**A**) image shows and enhancing tumor in the anterior horn of the left ventricle when she presented with first relapse of medulloblastoma. She achieved complete remission with salvage therapy; (**B**) image shows an enhancing tumor involving the anterior aspect and floor of the fourth ventricle (arrows) when she presented with second relapse; (**C**) image shows response to 3 months of metronomic therapy with decreased size and intensity of enhancing lesions; (**D**) image shows interval progression of the tumor seen in T2 FLAIR resulting in a change from temozolomide to etoposide at 4 months of treatment; (**E**) image shows resolution of the tumor mass and T2 FLAIR changes at 8 months of metronomic therapy; (**F**) image shows progressive disease at 10 months of treatment.

**Figure 4 jcm-11-02849-f004:**
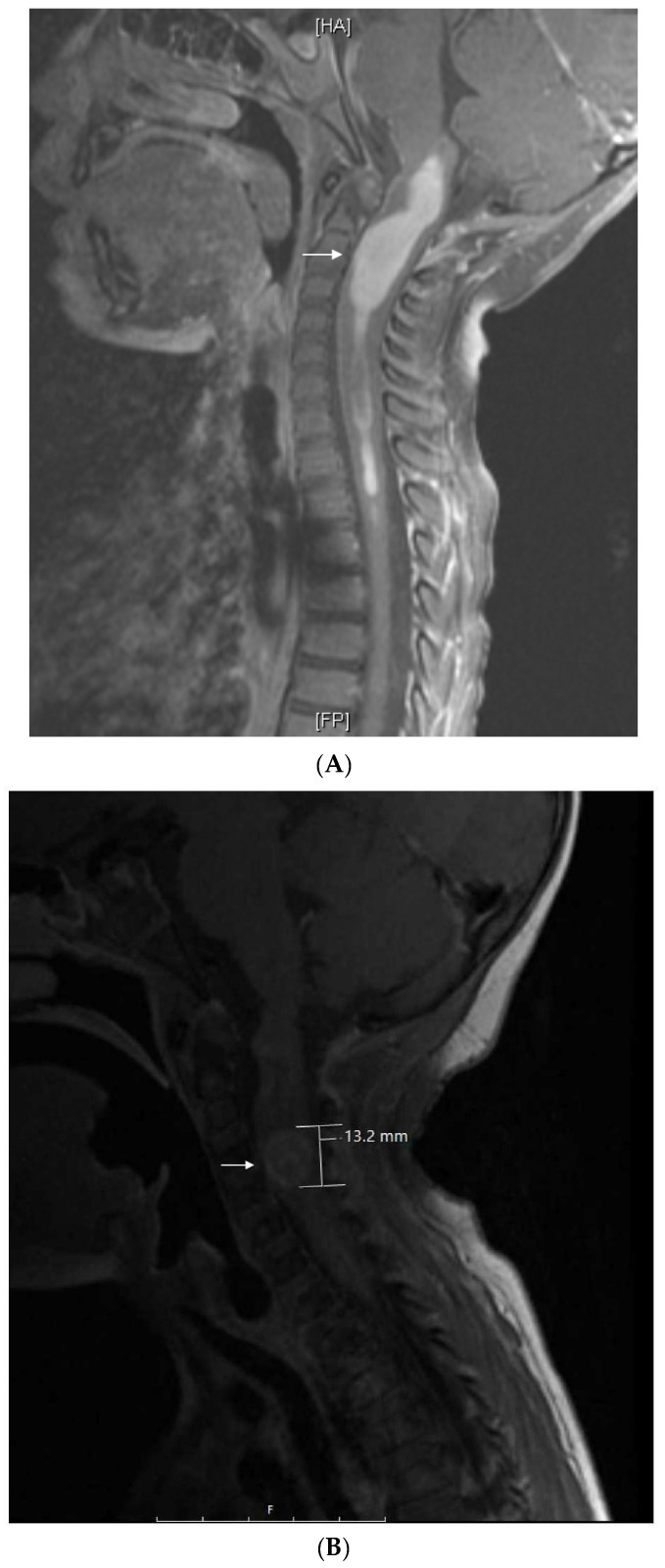
Patient 6: (**A**) MRI shows a 4.5 cm homogeneously enhancing expansile intramedullary tumor involving the medulla and upper cervical cord down to the level of C3–C4 (arrow) with an elongated syrinx extending inferiorly to the T3–T4 level; (**B**) the tumor was removed and treated with radiation but came back, for which it was treated with nine cycles of topotecan–ifosfamide–carboplatin with no significant change in tumor size (arrow) but significant toxicity, for which treatment was changed to metronomic chemotherapy; (**C**) after 3 months on metronomic chemotherapy, the patient recovered from toxicity and the tumor was slightly decreased; (**D**) the tumor was stable at the end of 4 years of metronomic chemotherapy.

**Figure 5 jcm-11-02849-f005:**
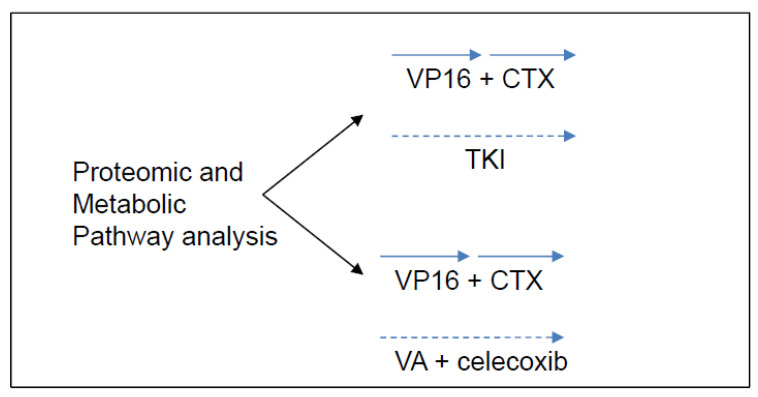
Schematic of future planned studies for pediatric cases eligible for metronomic chemotherapy. Proteomic and metabolic pathway analysis, where feasible, will be used to determine eligibility of patients to receive specific tyrosine kinase inhibitor (TKI) therapy coupled with metronomic chemotherapy, which involves alternating cycles of metronomic etoposide with metronomic cyclophosphamide. Patients not selected or not eligible for TKI-based therapies will receive valproic acid (VA) and celecoxib with metronomic chemotherapy as outlined in this manuscript.

**Table 1 jcm-11-02849-t001:** Summary of the 6 pediatric cases, treatment regimens, and clinical course.

Patient	Diagnosis	Regimen	Age at Initial Diagnosis	Best Response	Clinical Course
1	Rhabdoid tumor of the kidney (RTK) with lung and BM metastasis. Malignant myeloid clone at end of primary treatment. RTK relapse to bone marrow +/− lungs.	VP16-CTX, Celecoxib, VA	34 months	CR	Developed Treatment-related myeloid neoplasia and died from BMT complications in CR. CR 18 months at death. OS 3 years
2	Supratentorial Anaplastic Ependymoma. Local and distant relapse to spine.	VP16-CTX, Celecoxib, VA	6 years	CR	CR 5.5 years at last encounter Alive 8 years at last encounter
3	Medulloblastoma Infratentorial Microscopic leptomeningeal relapse	VP16 alone (VA added later)	10 months	CR	CR 5 years at last encounter Alive 7 years at last encounter
4	Medulloblastoma Infratentorial Second relapse	TMZ-CTX, VA, Celecoxib (TMZ changed to VP later)	5 years	PR	PR at 8 months lasted 2 months. Died of disease at 12 months
5	Metastatic neuroblastoma Retroperitoneal relapse	VP16-CTX, Sulindac (Sulindac changed to Celecoxib later)	9 years	PR	PR at 5 months lasted 6 months. Died of upper GI bleeding at 13 months.
6	Spinal cord neurocytoma Unresectable progressive tumor.	TMZ-CTX, VA, BV, Celecoxib	3 years	PR	PR at 3 months. Treatment stopped after 4 years of stable disease. PD 10 months later. She remains AWD on treatment

VP16 = etoposide, CTX = cyclophosphamide, VA = valproic acid, BV = bevacizumab, TMZ = temozolomide.

**Table 2 jcm-11-02849-t002:** Summary of proposed metronomic regimens.

Study	Metronomic Regimen
Kieran 2005 [10]	Thalidomide 3 mg/kg oral daily days 1–42,
	Celecoxib 100–400 mg bid oral days 1–42
	VP-16 50 mg/m^2^/day oral days 1–21
	CTX 2.5 mg/kg/day to a maximum of 100 mg oral days 22–42.
Kieran 2014 [9]	Celecoxib 100–400 mg bid oral days 1–42
	Thalidomide 3 mg/kg oral daily days 1–42,
	Fenofibrate 90 mg/m^2^ oral daily
	CTX 2.5 mg/kg/day to a max of 100 mg per day) days 1–21
	VP16 50 mg/m^2^/day days 22–42
Andre et al. 2008 [13]	VP-16 25 mg/m^2^/day days 1–14
	CTX 25 mg/m^2^/day days 15–28
	Celecoxib 100–400 mg/day days 1–28
MEMMAT Slavc et al. 2020 [39]	Etoposide 35–50 mg/m^2^/day oral on days 1–21 of 42 day cycles
	CTX 2.5 mg/kg/day oral on days 22–42 of 42 day cycles
	Intrathecal Liposomal cytarabine 16–30 mg days 1, 4, 8, 11 of 28-day cycles
	Intrathecal VP16 0.5 mg days 18, 19, 20, 21, 21 of 28-day cycles.
	Bevacizumab 10 mg/kg IV every other week
	Thalidomide 3 mg/kg daily 1 year
	Celecoxib 50–400 mg oral daily 1 year
	Fenofibrate 90 mg/m^2^ oral daily 1 year
Sun et al. 2021 [41]	VP16 25 mg/m^2^ oral days 1–21 of 56–day cycles
	Topotecan 1.4 mg/m^2^ oral daily on days 29–33 of 56-day cycles
	CTX 25–50 mg/m^2^ oral daily days 1–56 of 56-day cycles.
	Vinorelbine 40 mg/m^2^ oral weekly weeks 1–3 every 4 weeks
	Celecoxib 200 mg/m^2^ oral twice a day 1 year
This manuscript	VP-16 50 mg/m^2^/day days 1–21
	CTX 2.5 mg/kg/day days 22–42
	Celecoxib 250 mg/m^2^/dose twice a day days 1–42
	Valproic acid 7.5 mg/kg/dose twice a day days 1–42

## Data Availability

Not applicable.

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
