# Peer review of "Cyclic Metronomic Chemotherapy for Pediatric Tumors: Six Case Reports and a Review of the Literature"

_jcm, 2022, doi:10.3390/jcm11102849_

Round 1

Reviewer 1 Report

This revised draft is a significant improvement, and the authors’ replies and changes are appreciated.

Remaining issues to be addressed include:

  1. For patient 1, the authors should clarify the t(5;17) clone was not present at initial diagnosis, but developed after the receipt of standard chemotherapy and before the administration of metronomic therapy. Did the authors feel that the expansion of the t(5;17) clone was rhabdoid tumor, or myelodysplasia? This still is not clear from the text, and should be specifically addressed. Mention should be made that t(5;17) has been associated with therapy-related myelodysplasia and a reference should be provided. Also, the disease burden present at the time of starting metronomic chemotherapy consisted of two small lung nodules measuring 3 and 1.5 mm in size. These were presumed sites of tumor, and did they completely resolve?
  2. The death of patient 5 is attributed to upper GI bleeding. Was this bleeding related to tumor? If not, was there another plausible mechanism for this other than his prolonged exposure to sulindac and then celecoxib without use of an antacid? This draws into question the conclusions the authors make on page 15 lines 10-12, where they imply the only toxicities were hematologic and managed with dose reduction. This patient’s death and the potential link to NSAIDs must be better explained, and the authors should consider whether they agree with the recommendations in COG protocols that chronic use of NSAIDS should be accompanied by gastric prophylaxis with some type of antacid. If so, this should be more clearly stated. It is hard to have a patient in the series die from symptoms that are also known complications of NSAIDs and not more clearly identify this risk and strategies to mitigate this.
  3. Please review capitalization used in this manuscript, as many of the words that are capitalized such as drug names or disease names (e.g., Rhabdoid tumor) do not require capitalization.
  4. Please provide references for all protocol therapies listed (e.g, ACNS0121), as these may not be familiar to non-cooperative group members.
  5. The frequency of bevacizumab dosing for patient 6 should be reported.
  6. A colon should be used instead of a semicolon in the Abstract line 3.
  7. The figures and tables were not included in the revised version of the manuscript.

Reviewer 2 Report

The revised version of the manuscript by Carmaco and Francia still gives detailed description of 6 cases treated with different low-dose metronomic chemotherapy regimens and encouraging results. 

Although in the revised version the authors tried to adress my concerns, in the revised version the authors mainly report why the requested data is not available. 

Author Response

please see the attchment.
